# Re-Analyze Gauss: Bounds for Private Matrix Approximation via Dyson Brownian Motion

**Oren Mangoubi**
Worcester Polytechnic Institute

**Nisheeth K. Vishnoi**
Yale University

## Abstract

Given a symmetric matrix $M$ and a vector $\lambda$, we present new bounds on the Frobenius-distance utility of the Gaussian mechanism for approximating $M$ by a matrix whose spectrum is $\lambda$, under $(\varepsilon, \delta)$-differential privacy. Our bounds depend on both $\lambda$ and the gaps in the eigenvalues of $M$, and hold whenever the top $k + 1$ eigenvalues of $M$ have sufficiently large gaps. When applied to the problems of private rank-$k$ covariance matrix approximation and subspace recovery, our bounds yield improvements over previous bounds. Our bounds are obtained by viewing the addition of Gaussian noise as a continuous-time matrix Brownian motion. This viewpoint allows us to track the evolution of eigenvalues and eigenvectors of the matrix, which are governed by stochastic differential equations discovered by Dyson. These equations allow us to bound the utility as the square-root of a sum-of-squares of perturbations to the eigenvectors, as opposed to a sum of perturbation bounds obtained via Davis-Kahan-type theorems.

## 1 Introduction

Given a dataset $A \in \mathbb{R}^{m \times d}$, which consists of $m$ individuals with $d$-dimensional features, methods for preprocessing or prediction from $A$ often use the covariance matrix $M := A^\top A$ of $A$. In many such applications one computes a rank-$k$ approximation to $M$, or finds a matrix *close* to $M$ with a specified set of eigenvalues $\lambda = (\lambda_1, \dots, \lambda_d)$ [37, 28, 36]. Examples include the rank-$k$ covariance matrix approximation problem where one seeks to compute a rank-$k$ matrix which minimizes a given distance to $M$, and the subspace recovery problem where the goal is to compute a rank $k$-projection matrix $H = V_k V_k^\top$, where $V_k$ is the $d \times k$ matrix whose columns are the top-$k$ eigenvectors of $M$. These matrix approximation problems are ubiquitous in ML and have a rich algorithmic history; see [29, 45, 10, 8].

In some cases, the rows of $A$ correspond to sensitive features of individuals and the release of solutions to aforementioned matrix approximation problems may reveal their private information, e.g., as in the case of the Netflix prize problem [5]. Differential privacy (DP) has become a popular notion to quantify the extent to which an algorithm preserves privacy of individuals [15]. Algorithms for solving low-rank matrix approximation problems have been widely studied under DP constraints [30, 7, 19, 17]. Notions of DP studied in the literature include $(\varepsilon, \delta)$-DP [17, 25, 26, 19] which is the notion we study in this paper, as well as pure $(\varepsilon, 0)$-DP [17, 30, 2, 32]. To define a notion of DP in problems involving covariance matrices, following [7, 17], two matrices $M = A^\top A$ and $M' = A'^\top A'$ are said to be *neighbors* if they arise from $A, A'$ which differ by at most one row and as, is oftentimes done, require that each row of the datasets $A, A'$ has norm at most 1. For any $\varepsilon, \delta \geq 0$, a randomized mechanism $\mathcal{A}$ is $(\varepsilon, \delta)$-differentially private if for all neighbors $M, M' \in \mathbb{R}^{d \times d}$, and any measurable subset $S$ of outputs of $\mathcal{A}$, we have $\mathbb{P}(\mathcal{A}(M) \in S) \leq e^\varepsilon \mathbb{P}(\mathcal{A}(M') \in S) + \delta$.

36th Conference on Neural Information Processing Systems (NeurIPS 2022).

**The problem.** We consider a class of problems where one wishes to compute an approximation to a symmetric $d \times d$ matrix under $(\varepsilon, \delta)$-differential privacy constraints. Specifically, given $M = A^\top A$ for $A \in \mathbb{R}^{m \times d}$, together with a vector $\lambda$ of target eigenvalues $\lambda_1 \geq \cdots \geq \lambda_d$, the goal is to output a $d \times d$ matrix $\hat{H}$ with eigenvalues $\lambda$ which minimizes the Frobenius-norm distance $\|\hat{H} - H\|_F$ under $(\varepsilon, \delta)$-differential privacy constraints. Here $H$ is the matrix with eigenvalues $\lambda$ and the same eigenvectors as $M$. This class of problems includes as a special case the subspace recovery problem if we set $\lambda_1 = \cdots = \lambda_k = 1$ and $\lambda_{k+1} = \cdots = \lambda_d = 0$. It also includes the rank-$k$ covariance approximation problems if we set $\lambda_i = \sigma_i$ for $i \leq k$, where $\sigma_1 \geq \cdots \geq \sigma_d$ are the eigenvalues of $M$. Since revealing $\sigma_i$s may violate privacy constraints, the eigenvalues of the output matrix $\hat{H}$ should not be the same as those of $H$.

Various distance functions have been used in the literature to evaluate the utility of $(\varepsilon, \delta)$-DP mechanisms for matrix approximation problems, including the Frobenius-norm distance $\|\hat{H} - H\|_F$ (e.g. [19, 2])and the Frobenius inner product utility $\langle M, H - \hat{H} \rangle$ (e.g. [11, 19, 24]). Note that while a bound $\|H - \hat{H}\|_F \leq b$ implies an upper bound on the inner product utility of $\langle M, H - \hat{H} \rangle \leq \|M\|_F \cdot b$ (by the Cauchy-Schwarz inequality), an upper bound on the inner product utility does not (in general) imply any upper bound on the Frobenius-norm distance. Moreover, the Frobenius-norm distance can be a good utility metric to use if the goal is to recover a low rank matrix $H$ from a dataset of noisy observations (see e.g. [12]). Hence, we use the Frobenius-norm distance to measure the utility of an $(\varepsilon, \delta)$-DP mechanism.

**Related work.** The problem of approximating a matrix under differential privacy constraints has been widely studied. In particular, prior works have provided algorithms for problems where the goal is to approximate a covariance matrix under differential privacy constraints, including rank-$k$ PCA and subspace recovery [7, 30, 19, 33] as well as rank-$k$ covariance matrix approximation [7, 19, 2]. Another set of works have studied the problem of approximating a rectangular data matrix $A$ under DP [7, 1, 25, 26]. We note that upper bounds on the utility of differentially-private mechanisms for rectangular matrix approximation problems can grow with the number of datapoints $m$, while those for covariance matrix approximation problems oftentimes depend only on the dimension $d$ of the covariance matrix and do not grow with $m$. Prior works which deal with covariance matrix approximation problems such as rank-$k$ covariance matrix approximation and subspace recovery are the most relevant to our paper. The notion of DP varies among the different works on differentially-private matrix approximation, with many of these works considering the notion $(\varepsilon, \delta)$-DP [25, 26, 19], while other works focus on (pure) $(\varepsilon, 0)$-DP [30, 2, 33].

*Analysis of the Gaussian mechanism in [19].* [19] analyze a version of the Gaussian mechanism of [16], where one perturbs the entries of $M$ by adding a symmetric matrix $E$ with i.i.d. Gaussian entries $N(0, \sqrt{\log(\frac{1}{\delta})}/\varepsilon)$, to obtain an $(\varepsilon, \delta)$-differentially private mechanism which outputs a perturbed matrix $\hat{M} = M + E$. One can then post-process this matrix $\hat{M}$ to obtain a rank-$k$ projection matrix which projects onto the subspace spanned by the top-$k$ eigenvectors of $\hat{M}$ (for the rank-$k$ PCA or subspace recovery problem), or a rank-$k$ matrix $\hat{H}$ with the same top-$k$ eigenvectors and eigenvalues as $\hat{M}$ (for the rank-$k$ covariance matrix approximation problem). [19] consider different notions of utility in their results, including the inner product utility (for PCA), and the Frobenius-norm and spectral-norm distance distances (for low-rank approximation and subspace recovery).

In one set of results, [19] give lower utility bounds of $\tilde{\Omega}(k\sqrt{d})$ w.h.p. for the rank-$k$ PCA problem with respect to the inner product utility $\langle M, H \rangle$, together with matching upper bounds provided by a post-processing of the Gaussian mechanism, where $\tilde{\Omega}$ hides polynomial factors of $\frac{1}{\varepsilon}$ and $\log(\frac{1}{\delta})$ (their Theorems 3 and 18). As noted by the authors, their lower bounds are tight for matrices $M$ with the "worst-case" spectral profile $\sigma$, but they can obtain improved upper bounds for matrices $M$ where $\sigma_k - \sigma_{k+1} > \tilde{\Omega}(\sqrt{d})$ (Theorem 3 of [19]).

For the subspace recovery problem, [19] obtain a Frobenius-distance bound of $\|\hat{H} - H\|_F \leq \tilde{O}\left(\sqrt{kd}/(\sigma_k - \sigma_{k+1})\right)$ w.h.p. for a post-processing of the Gaussian mechanism whenever $\sigma_k - \sigma_{k+1} > \tilde{\Omega}(\sqrt{d})$ (implied by their Theorem 6, which is stated for the spectral norm). And for the rank-$k$ covariance matrix approximation problem, [19] show a utility bound of

$\|\hat{H} - M\|_F - \|H - M\|_F \leq \tilde{O}(k\sqrt{d})$ w.h.p. for a post-processing of the Gaussian mechanism (Theorem 7 in [19]), and also give related bounds for the spectral norm. While their Frobenius bound for the covariance matrix approximation problem is independent of the number of datapoints $m$, it may not be tight. For instance, when $k = d$, one can easily obtain a better bound since, by the triangle inequality, $\|\hat{H} - M\|_F - \|H - M\|_F \leq \|\hat{H} - H\|_F = \|\hat{M} - M\|_F = \|E\|_F \leq O(d)$ w.h.p., since $\|E\|_F$ is just the norm of a vector of $d^2$ Gaussians with variance $\tilde{O}(1)$. Moreover, the bound for the the rank-$k$ covariance approximation problem, $\|\hat{H} - H\|_F \leq \tilde{O}(k\sqrt{d})$, is also a worst-case upper bound for any spectral profile $\sigma$ as the right hand side of the bound not depend on the eigenvalues $\sigma$.

*Thus, a question arises of whether the Frobenius-norm utility bounds for the rank-k covariance matrix approximation and subspace recovery problems are tight for all spectral profiles $\sigma$, and whether the analysis of the Gaussian mechanism can be improved to achieve better utility bounds. A more general question is to obtain utility bounds for the Gaussian mechanism for the matrix approximation problems for arbitrary $\lambda$.*

**Our contribution.** Our main result is a new upper bound on the Frobenius-distance utility of the Gaussian mechanism for the general matrix approximation problem for a given $M$ and $\lambda$ (Theorem 2.2). Our bound depends on the eigenvalues of $M$ and the entries of $\lambda$.

The novel insight is to view the perturbed matrix $M + E$ as a continuous-time symmetric matrix diffusion, where each entry of the matrix $M + E$ is the value reached by a (one-dimensional) Brownian motion after some time $T = \sqrt{\log(\frac{1}{\delta})}/\varepsilon$. This matrix-valued Brownian motion, which we denote by $\Phi(t)$, induces a stochastic process on the eigenvalues $\gamma_1(t) \geq \cdots \geq \gamma_d(t)$ and corresponding eigenvectors $u_1(t), \ldots, u_d(t)$ of $\Phi(t)$ originally discovered by Dyson and now referred to as Dyson Brownian motion, with initial values $\gamma_i(0) = \sigma_i$ and $u_i(0)$ which are the eigenvalues and eigenvectors of the initial matrix $M$ [20].

We then use the stochastic differential equations (3) and (4), which govern the evolution of the eigenvalues and eigenvectors of the Dyson Brownian motion, to track the perturbations to each eigenvector. Roughly speaking, these equations say that, as the Dyson Brownian motion evolves over time, every pair of eigenvalues $\gamma_i(t)$ and $\gamma_j(t)$, and corresponding eigenvectors $u_i(t)$ and $u_j(t)$, interacts with the other eigenvalue/eigenvector with the magnitude of the interaction term proportional to $\frac{1}{\gamma_i(t) - \gamma_j(t)}$ at any given time $t$. This allows us to bound the perturbation of the eigenvectors at every time $t$, provided that the initial gaps in the top $k+1$ eigenvalues of the input matrix are $\geq \Omega(\sqrt{d})$ (Assumption 2.1). Empirically, we observe that Assumption 2.1 is satisfied for covariance matrices of many real-world datasets (see Appendix J), as well as on Wishart random matrices $W = A^\top A$, where $A$ is an $m \times d$ matrix of i.i.d. Gaussian entries, for sufficiently large $m$ (see Appendix I). We then derive a stochastic differential equation which tracks how the utility changes as the Dyson Brownian motion evolves over time (Lemma 4.1), and integrate this differential equation over time to obtain a bound on the (expectation of) the utility $\mathbb{E}[\|\hat{H} - H\|_F]$ (Lemma 4.5) as a function of the gaps $\gamma_i(t) - \gamma_j(t)$.

Plugging in basic estimates (Lemma 4.4) for the eigenvalue gaps $\gamma_i(t) - \gamma_j(t)$ to Lemma 4.5, we obtain a bound on the expected utility $\mathbb{E}[\|\hat{H} - H\|_F]$ (Theorem 2.2) for the different matrix approximation problems as a function of the eigenvalue gaps $\sigma_i - \sigma_j$ of the input matrix $M$. Roughly speaking, our bound is the square-root of a sum-of-squares of the ratios, $\frac{\lambda_i - \lambda_j}{\sigma_i - \max(\sigma_j, \sigma_{k+1})}$, of eigenvalue gaps of the input and output matrices.

When applied to the rank-$k$ covariance matrix approximation problem (Corollary 2.3), Theorem 2.2 implies a bound of $\mathbb{E}[\|\hat{H} - H\|_F] \leq \tilde{O}(\sqrt{kd})$ whenever the eigenvalues $\sigma$ of the input matrix $M$ satisfy $\sigma_k - \sigma_{k+1} \geq \Omega(\sigma_k)$ and the gaps in top $k + 1$ eigenvalues satisfy $\sigma_i - \sigma_{i+1} \geq \tilde{\Omega}(\sqrt{d})$. Thus, when $M$ satisfies the above condition on $\sigma$, our bound improves by a factor of $\sqrt{k}$ on the (expectation of) the previous bound of [19], which says that $\|\hat{H} - M\|_F - \|H - M\|_F \leq \tilde{O}(k\sqrt{d})$ w.h.p., since by the triangle inequality $\|\hat{H} - M\|_F - \|H - M\|_F \leq \|\hat{H} - H\|_F$. This condition on $\sigma$ is satisfied, e.g., for matrices $M$ whose eigenvalue gaps are at least as large as those of the Wishart random covariance matrices with sufficiently many datapoints $m$ (see Section 2 for details). And,

if $\sigma$ is such that $\sigma_i - \sigma_{i+1} \geq \Omega(\sigma_k - \sigma_{k+1})$ for $i \leq k$, Theorem 2.2 implies a bound of $\mathbb{E}[\|\hat{H} - H\|_F] \leq \tilde{O}(\sqrt{d}/(\sigma_k - \sigma_{k+1}))$ for the subspace recovery problem (Corollary 2.4), improving by a factor of $\sqrt{k}$ (in expectation) on the previous bound of [19], which implies that $\|\hat{H} - M\|_F - \|H - M\|_F \leq \tilde{O}\left(\sqrt{kd}/(\sigma_k - \sigma_{k+1})\right)$ w.h.p.

## 2  Results

Our main result (Theorem 2.2) gives a new and unified upper bound on the Frobenius-norm utility of a post-processing of the Gaussian mechanism, for the general matrix approximation problem where one is given a symmetric matrix $M \in \mathbb{R}^{d \times d}$ and a vector $\lambda$ with $\lambda_1 \geq \cdots \geq \lambda_d$, and the goal is to compute a matrix $\hat{H}$ with eigenvalues $\lambda$ which minimizes the distance $\|\hat{H} - H\|_F$. Here $H$ is the matrix with eigenvalues $\lambda$ and the same eigenvectors as $M$. Plugging in different choices of $\lambda$ to Theorem 2.2, we obtain as corollaries new Frobenius-distance utility bounds for the rank-$k$ covariance matrix approximation problem (Corollary 2.3) and the subspace recovery problem (Corollary 2.4). Our results rely on the following assumption about the eigenvalues of the input matrix $M$:

**Assumption 2.1** (($M, k, \lambda_1, \varepsilon, \delta$) Eigenvalue gaps)**.** *The gaps in the top $k + 1$ eigenvalues eigenvalues $\sigma_1 \geq \cdots \geq \sigma_d$ of the matrix $M \in \mathbb{R}^{d \times d}$ satisfy $\sigma_i - \sigma_{i+1} \geq \frac{8\sqrt{\log(\frac{1.25}{\delta})}}{\varepsilon}\sqrt{d} + 3\log^{\frac{1}{2}}(\lambda_1 k)$ for every $i \in [k]$.*

We observe empirically that Assumption 2.1 is satisfied on a number of real-world datasets which were previously used as benchmarks in the differentially private matrix approximation literature [11, 2] (see Appendix J). Assumption 2.1 is also satisfied, for instance, by random Wishart matrices $W = A^\top A$, where $A$ is an $m \times d$ matrix of i.i.d. Gaussian entries, which are a popular model for sample covariance matrices [47]. This is because the minimum gap $\sigma_i - \sigma_{i+1}$ of a Wishart matrix grows proportional to $\sqrt{m}$ with high probability; thus for large enough $m$, Assumption 2.1 holds (see Appendix I for details). Hence, the assumption requires that the gaps in the top $k + 1$ eigenvalues of $M$ are at least as large as the gaps in a random Wishart matrix.

**Theorem 2.2** (**Main result**)**.** *Let $\varepsilon, \delta > 0$, and given a symmetric matrix $M \in \mathbb{R}^{d \times d}$ with eigenvalues $\sigma_1 \geq \cdots \geq \sigma_d$ and corresponding orthonormal eigenvectors $v_1, \ldots, v_d$. Let $G$ be a matrix with i.i.d. $N(0, 1)$ entries, and consider the mechanism that outputs $\hat{M} = M + \frac{\sqrt{2\log(\frac{1.25}{\delta})}}{\varepsilon}(G + G^\top)$. Then such a mechanism is $(\varepsilon, \delta)$-differentially private. Moreover, let $\lambda_1 \geq \cdots \geq \lambda_d$ and $k \in [d]$ be any numbers such that $\lambda_i = 0$ for $i > k$, and define $\Lambda := \mathrm{diag}(\lambda_1, \ldots, \lambda_d)$ and $V = [v_1, \ldots, v_d]$, and define $\hat{\sigma}_1 \geq \cdots \geq \hat{\sigma}_d$ to be the eigenvalues of $\hat{M}$ with corresponding orthonormal eigenvectors $\hat{v}_1, \ldots, \hat{v}_d$ and $\hat{V} = [\hat{v}_1, \ldots, \hat{v}_d]$. Then if $M$ satisfies Assumption 2.1 for $(M, k, \lambda_1, \varepsilon, \delta)$, we have*

$$\mathbb{E}\left[\|\hat{V}\Lambda\hat{V}^\top - V\Lambda V^\top\|_F^2\right] \leq O\left(\sum_{i=1}^{k}\sum_{j=i+1}^{d}\frac{(\lambda_i - \lambda_j)^2}{(\sigma_i - \max(\sigma_j, \sigma_{k+1}))^2}\right)\frac{\log(\frac{1}{\delta})}{\varepsilon^2}$$

The fact that the mechanism in this theorem is $(\varepsilon, \delta)$-differentially private follows from standard results about the Gaussian mechanism [19]. Given any list of eigenvalues $\lambda$, and letting $\Lambda = \mathrm{diag}(\lambda)$, one can post-process the matrix $\hat{M}$ by computing its spectral decomposition $\hat{M} = \hat{V}\hat{\Sigma}\hat{V}^\top$ and replacing its eigenvalues to obtain a matrix $\hat{V}\Lambda\hat{V}^\top$ with eigenvalues $\lambda$ and eigenvectors $\hat{V}$. Since $\hat{V}\Lambda\hat{V}^\top$ is a post-processing of the Gaussian mechanism, the mechanism which outputs $\hat{V}\Lambda\hat{V}^\top$ is differentially private as well. Theorem 2.2 bounds the excess utility $\mathbb{E}[\|\hat{V}\Lambda\hat{V}^\top - V\Lambda V^\top\|_F^2]$ (whenever the gaps in the eigenvalues $\sigma_1 \geq \cdots \geq \sigma_d$ of the input matrix satisfy Assumption 2.1) as a sum-of-squares of the ratio of the gaps $\lambda_i - \lambda_j$ in the given eigenvalues to the corresponding gaps $\sigma_i - \max(\sigma_j, \sigma_{k+1})$ in the eigenvalues of the input matrix (note that $\lambda_i - \lambda_j = \lambda_i - \max(\lambda_j, \lambda_{k+1})$ since $\lambda_j = 0$ for $j \geq k + 1$).

While we do not know if Theorem 2.2 is tight for all choices of $\lambda$ and $k$, it does give a tight bound for some problems. Namely, when applied to the covariance matrix estimation

problem, in the special case where $k = d$ Theorem 2.2 implies a bound of $\mathbb{E}[\|\hat{M} - M\|_F] \leq \tilde{O}(\sqrt{kd}) = O(d)$ (see Corollary 2.3). Since $\hat{M} - M = \frac{\sqrt{2\log(\frac{1.25}{\delta})}}{\varepsilon}(G + G^\top)$, the matrix $\hat{M} - M$ has independent Gaussian entries with mean zero and variance $\tilde{O}(1)$, and we have from concentration results for Gaussian random matrices (see e.g. Theorem 2.3.6 of [39]) that $\mathbb{E}[\|\hat{M} - M\|_F] = \tilde{\Omega}(d)$, implying that the bound in Theorem 2.2 is tight in this case.

The proof of Theorem 2.2 differs from prior works, including that of [19] which use Davis-Kahan-type theorems [13] and trace inequalities, and instead relies on an interpretation of the Gaussian mechanism as a diffusion process which may be of independent interest (See Appendix K for additional comparison to previous approaches). This connection allows us to use sophisticated tools from stochastic differential equations and random matrix theory. We present an outline of the proof in Section 4.

**Application to covariance matrix approximation:** Plugging $\lambda_i = \sigma_i$ for $i \leq k$ and $\lambda_i = 0$ for $i > k$ into Theorem 2.2, and plugging in concentration bounds for the perturbation to the eigenvalues $\sigma_i$, we obtain utility bounds for covariance matrix approximation:

**Corollary 2.3** (**Rank-$k$ covariance matrix approximation**). *Let $\varepsilon, \delta > 0$, and given a symmetric matrix $M \in \mathbb{R}^{d \times d}$ with eigenvalues $\sigma_1 \geq \cdots \geq \sigma_d$ and corresponding orthonormal eigenvectors $v_1, \ldots, v_d$. Let $G$ be a matrix with i.i.d. $N(0,1)$ entries, and consider the mechanism that outputs $\hat{M} = M + \frac{\sqrt{2\log(\frac{1.25}{\delta})}}{\varepsilon}(G + G^\top)$. Then such a mechanism is $(\varepsilon, \delta)$-differentially private. Moreover, for any $k \in [d]$, define $\Sigma_k := \text{diag}(\sigma_1, \ldots, \sigma_k, 0 \ldots, 0)$ and $V = [v_1, \ldots, v_d]$, and define $\hat{\sigma}_1 \geq \cdots \geq \hat{\sigma}_d$ to be the eigenvalues of $\hat{M}$ with corresponding orthonormal eigenvectors $\hat{v}_1, \ldots, \hat{v}_d$, and define $\hat{\Sigma}_k := \text{diag}(\hat{\sigma}_1, \ldots, \hat{\sigma}_k, 0 \ldots, 0)$ and $\hat{V} := [\hat{v}_1, \ldots, \hat{v}_d]$. Then if $M$ satisfies Assumption 2.1 for $(M, k, \sigma_1, \varepsilon, \delta)$, and defining $\sigma_{d+1} := 0$, we have*

$$\mathbb{E}\left[\|\hat{V}\hat{\Sigma}_k\hat{V}^\top - V\Sigma_k V^\top\|_F\right] \leq O\left(\sqrt{kd} \times \frac{\sigma_k}{\sigma_k - \sigma_{k+1}}\right)\frac{\log^{\frac{1}{2}}(\frac{1}{\delta})}{\varepsilon}.$$

The proof appears in Appendix G. If $\sigma_k - \sigma_{k+1} = \Omega(\sigma_k)$, then Corollary 2.3 implies that $\mathbb{E}\left[\|\hat{V}\hat{\Sigma}_k\hat{V}^\top - V\Sigma_k V^\top\|_F\right] \leq O\left(\sqrt{kd}\frac{\log^{\frac{1}{2}}(\frac{1}{\delta})}{\varepsilon}\right)$. Thus, for matrices $M$ with eigenvalues satisfying Assumption 2.1 and where $\sigma_k - \sigma_{k+1} = \Omega(\sigma_k)$, Corollary 2.3 improves by a factor of $\sqrt{k}$ on the bound in Theorem 7 of [19] which says $\|\hat{V}\hat{\Sigma}_k\hat{V}^\top - M\|_F - \|V\Sigma_k V^\top - M\|_F = \tilde{O}(k\sqrt{d})$ w.h.p.. This is because an upper bound on $\|\hat{V}\hat{\Sigma}_k\hat{V}^\top - V\Sigma_k V^\top\|_F$ implies an upper bound on $\|\hat{V}\hat{\Sigma}_k\hat{V}^\top - M\|_F - \|V\Sigma_k V^\top - M\|_F$ by the triangle inequality. On the other hand, while their result does not require a bound on the gaps in the eigenvalue of $M$ and bounds their utility w.h.p., our Corollary 2.4 requires a bound on the gaps of the top $k+1$ eigenvalues of $M$ and bounds the expected utility $\mathbb{E}[\|\hat{V}\hat{\Sigma}_k\hat{V}^\top - V\Sigma_k V^\top\|_F]$.

**Application to subspace recovery:** Plugging in $\lambda_1 = \cdots = \lambda_k = 1$ and $\lambda_{k+1} = \cdots = \lambda_d = 0$, the post-processing step in Theorem 2.2 outputs a projection matrix, and we obtain utility bounds for the subspace recovery problem.

**Corollary 2.4** (**Subspace recovery**). *Let $\varepsilon, \delta > 0$, and given a symmetric matrix $M \in \mathbb{R}^{d \times d}$ with eigenvalues $\sigma_1 \geq \cdots \geq \sigma_d$ and corresponding orthonormal eigenvectors $v_1, \ldots, v_d$. Let $G$ be a matrix with i.i.d. $N(0,1)$ entries, and consider the mechanism that outputs $\hat{M} = M + \frac{\sqrt{2\log(\frac{1.25}{\delta})}}{\varepsilon}(G + G^\top)$. Then such a mechanism is $(\varepsilon, \delta)$-differentially private. Moreover, for any $k \in [d]$, define the $d \times k$ matrices $V_k = [v_1, \ldots, v_k]$ and $\hat{V}_k = [\hat{v}_1, \ldots, \hat{v}_k]$, where $\hat{\sigma}_1 \geq \cdots \geq \hat{\sigma}_d$ denote the eigenvalues of $\hat{M}$ with corresponding orthonormal eigenvectors $\hat{v}_1, \ldots, \hat{v}_d$. Then if $M$ satisfies Assumption 2.1 for $(M, k, 2, \varepsilon, \delta)$, we have $\mathbb{E}\left[\|\hat{V}_k\hat{V}_k^\top - V_k V_k^\top\|_F\right] \leq O\left(\frac{\sqrt{kd}}{\sigma_k - \sigma_{k+1}} \times \frac{\log^{\frac{1}{2}}(\frac{1}{\delta})}{\varepsilon}\right)$. Moreover, if we also have that $\sigma_i - \sigma_{i+1} \geq \Omega(\sigma_k - \sigma_{k+1})$ for all $i \leq k$, then*

$$\mathbb{E}\left[\|\hat{V}_k\hat{V}_k^\top - V_k V_k^\top\|_F\right] \leq O\left(\frac{\sqrt{d}}{\sigma_k - \sigma_{k+1}} \times \frac{\log^{\frac{1}{2}}(\frac{1}{\delta})}{\varepsilon}\right).$$

The proof appears in Appendix H. For matrices $M$ satisfying Assumption 2.1, the first inequality of Corollary 2.4 recovers (in expectation) the Frobenius-norm utility bound implied by Theorem 6 of [19], which states that $\|\hat{V}_k \hat{V}_k^\top - V_k V_k^\top\|_F \leq O\left(\frac{\sqrt{kd}}{\sigma_k - \sigma_{k+1}} \times \frac{\log^{\frac{1}{2}}(\frac{1}{\delta})}{\varepsilon}\right)$ w.h.p. Moreover, for many input matrices $M$ with spectral profiles $\sigma_1 \geq \cdots \geq \sigma_d$ satisfying Assumption 2.1, Theorem 2.2 implies stronger bounds than those in [19] for the subspace recovery problem. For instance, if we also have that $\sigma_i - \sigma_{i+1} \geq \Omega(\sigma_k - \sigma_{k+1})$ for all $i \leq k$, the bound given in the second inequality of Corollary 2.4 improves on the bound of [19] by a factor of $\sqrt{k}$. On the other hand, while their result only requires that $\sigma_k - \sigma_{k+1} \geq \sqrt{d}$ and bounds the Frobenius distance $\|\hat{V}_k \hat{V}_k^\top - V_k V_k^\top\|_F$ w.h.p., our Corollary 2.4 requires a bound on the gaps of the top $k + 1$ eigenvalues of $M$ and bounds the expected Frobenius distance $\mathbb{E}[\|\hat{V}_k \hat{V}_k^\top - V_k V_k^\top\|_F]$.

## 3    Preliminaries

**Brownian motion and stochastic calculus.** A Brownian motion $W(t)$ in $\mathbb{R}$ is a continuous process that has stationary independent increments (see e.g., [34]). In a multi-dimensional Brownian motion, each coordinate is an independent and identical Brownian motion. The filtration $\mathcal{F}_t$ generated by $W(t)$ is defined as $\sigma\left(\cup_{s \leq t} \sigma(W(s))\right)$, where $\sigma(\Omega)$ is the $\sigma$-algebra generated by $\Omega$. $W(t)$ is a martingale with respect to $\mathcal{F}_t$.

**Definition 3.1** (**Itô Integral**). *Let $W(t)$ be a Brownian motion for $t \geq 0$, let $\mathcal{F}_t$ be the filtration generated by $W(t)$, and let $z(t) : \mathcal{F}_t \to \mathbb{R}$ be a stochastic process adapted to $\mathcal{F}_t$. The Itô integral is defined as $\int_0^T z(t)\mathrm{d}W(t) := \lim_{\omega \to 0} \sum_{i=1}^{\frac{T}{\omega}} z(i\omega) \times [W((i+1)\omega) - W(i\omega)]$.*

**Lemma 3.1** (**Itô's Lemma, integral form with no drift; Theorem 3.7.1 of [31]**). *Let $f : \mathbb{R}^n \to \mathbb{R}$ be any twice-differentiable function. Let $W(t) \in \mathbb{R}^n$ be a Brownian motion, and let $X(t) \in \mathbb{R}^n$ be an Itô diffusion process with mean zero defined by the following stochastic differential equation:*

$$\mathrm{d}X_j(t) = \sum_{i=1}^d R_{ij}(t)\mathrm{d}W_i(t), \tag{1}$$

*for some Itô diffusion $R(t) \in \mathbb{R}^{n \times n}$ adapted to the filtration generated by the Brownian motion $W(t)$. Then for any $T \geq 0$,*

$$f(X(T)) - f(X(0)) = \int_0^T \sum_{i=1}^n \sum_{\ell=1}^n \left(\frac{\partial}{\partial X_\ell} f(X(t))\right) R_{i\ell}(t)\mathrm{d}W_i(t)$$

$$+ \frac{1}{2} \int_0^T \sum_{i=1}^n \sum_{j=1}^n \sum_{\ell=1}^n \left(\frac{\partial^2}{\partial X_j \partial X_\ell} f(X(t))\right) R_{ij}(t) R_{i\ell}(t)\mathrm{d}t.$$

**Dyson Brownian motion.** Let $W(t) \in \mathbb{R}^{d \times d}$ be a matrix where each entry is an independent standard Brownian motion with distribution $N(0, tI_d)$ at time $t$, and let $B(t) = W(t) + W^\top(t)$. Define the symmetric-matrix valued stochastic process $\Phi(t)$ as follows:

$$\Phi(t) := M + B(t) \qquad \forall t \geq 0. \tag{2}$$

The process $\Phi(t)$ is referred to as (matrix) Dyson Brownian motion. At every time $t > 0$ the eigenvalues $\gamma_1(t), \ldots, \gamma_d(t)$ of $\Phi(t)$ are distinct with probability 1, and (2) induces a stochastic process on the eigenvalues and eigenvectors. The process on the eigenvalues and eigenvectors can be expressed via the following diffusion equations. The eigenvalue diffusion process, which is also referred to as (eigenvalue) "Dyson Brownian motion", is defined by the stochastic differential equation (3). The (eigenvalue) Dyson Brownian motion is an Itô diffusion and can be expressed can be expressed by the following stochastic differential equation [20]:

$$\mathrm{d}\gamma_i(t) = \mathrm{d}B_{ii}(t) + \sum_{j \neq i} \frac{1}{\gamma_i(t) - \gamma_j(t)}\mathrm{d}t \qquad \forall i \in [d], t > 0. \tag{3}$$

The corresponding eigenvector process $v_1(t), \dots, v_d(t)$, referred to as the Dyson vector flow, is also an Itô diffusion and, conditional on the eigenvalue process (3), can be expressed by the following stochastic differential equation (see e.g., [3]):

$$\mathrm{d}u_i(t) = \sum_{j \neq i} \frac{\mathrm{d}B_{ij}(t)}{\gamma_i(t) - \gamma_j(t)} u_j(t) - \frac{1}{2} \sum_{j \neq i} \frac{\mathrm{d}t}{(\gamma_i(t) - \gamma_j(t))^2} u_i(t) \qquad \forall i \in [d], t > 0. \quad (4)$$

**Eigenvalue bounds.** The following two Lemmas will help us bound the gaps in the eigenvalues of the Dyson Brownian motion:

**Lemma 3.2** (Theorem 4.4.5 of [43], special case [1]). *Let $W \in \mathbb{R}^{d \times d}$ with i.i.d. $N(0,1)$ entries. Then $\mathbb{P}(\|W\|_2 > 2(\sqrt{d} + s)) < 2e^{-s^2}$ for any $s > 0$.*

**Lemma 3.3** (**Weyl's Inequality; [6]**). *If $A, B \in \mathbb{R}^{d \times d}$ are two symmetric matrices, and denoting the $i$'th-largest eigenvalue of any symmetric matrix $M$ by $\sigma_i(M)$, we have $\sigma_i(A) + \sigma_d(B) \leq \sigma_i(A + B) \leq \sigma_i(A) + \sigma_1(B)$.*

# 4 Proof of Theorem 2.2

We give an overview of the proof of Theorem 2.2, along with the main technical lemmas used to prove this result. Section 4.1 outlines the different steps in our proof. In Steps 1 and 2 we construct the matrix-valued diffusion used in our proof. Steps 3,4, and 5 present the main technical lemmas, and in step 6 we explain how to complete the proof. The statements of the lemmas and the highlights of their proofs, are given in Sections 4.2, 4.3, 4.4. In section 4.5 we explain how to complete the proof. The full proofs are deferred to the appendix.

## 4.1 Outline of proof

1. **Step 1: Expressing the Gaussian Mechanism as a Dyson Brownian Motion.** To obtain our utility bound, we view the Gaussian mechanism as a matrix-valued Brownian motion (2) initialized at the input matrix $M$: $\Phi(t) := M + B(t) \qquad \forall t \geq 0$. If we run this Brownian motion for time $T = \sqrt{2 \log(\frac{1.25}{\delta})}/\varepsilon$ we have that $\Phi(T) = (\sqrt{2 \log(\frac{1.25}{\delta})}/\varepsilon)(G + G^\top)$, recovering the output of the Gaussian mechanism. In other words, the input to the Gaussian mechanism is $M = \Phi(0)$, and the output is $\hat{M} = \Phi(T)$.

2. **Step 2: Expressing the post-processed mechanism as a matrix diffusion $\Psi(t)$.** Our goal is to bound $\|\hat{V}\Lambda\hat{V}^\top - V\Lambda V^\top\|_F$, where $M = V\Sigma V^\top$ and $\hat{M} = \hat{V}\hat{\Sigma}\hat{V}^\top$ are spectral decompositions of $M$ and $\hat{M}$. To bound the error $\|\hat{V}\Lambda\hat{V}^\top - V\Lambda V^\top\|_F$ we will define a stochastic process $\Psi(t)$ such that $\Psi(0) = V\Lambda V^\top$ and $\Psi(t) = \hat{V}\Lambda\hat{V}^\top$, and then bound the Frobenius distance $\|\Psi(T) - \Psi(0)\|_F$ by integrating the (stochastic) derivative of $\Psi(t)$ over the time interval $[0, T]$.

   Towards this end, at every time $t$, let $\Phi(t) = U(t)\Gamma(t)U(t)^\top$ be a spectral decomposition of the symmetric matrix $\Phi(t)$, where $\Gamma(t)$ is a diagonal matrix with diagonal entries $\gamma_1(t) \geq \dots \geq \gamma_d(t)$ that are the eigenvalues of $\Phi(t)$, and $U(t) = [u_1(t), \dots, u_d(t)]$ is a $d \times d$ orthogonal matrix whose columns $u_1(t), \dots, u_d(t)$ are an orthonormal basis of eigenvectors of $\Phi(t)$. At every time $t$, define $\Psi(t)$ to be the symmetric matrix with eigenvalues $\Lambda$ and eigenvectors given by the columns of $U(t)$: $\Psi(t) := U(t)\Lambda U(t)^\top \ \forall t \in [0, T]$.

3. **Step 3: Computing the stochastic derivative $\mathrm{d}\Psi(t)$.** To bound the expected squared Frobenius distance $\mathbb{E}[\|\Psi(T) - \Psi(0)\|_F^2]$, we first compute the stochastic derivative $\mathrm{d}\Psi(t)$ of the matrix diffusion $\Psi(T)$ (Lemma 4.2).

4. **Step 4: Bounding the eigenvalue gaps.** The equation for the derivative $\mathrm{d}\Psi(t)$ includes terms with magnitude proportional to the inverse of the eigenvalue gaps $\Delta_{ij}(t) := \gamma_i(t) - \gamma_j(t)$ for each $i, j \in [d]$, which evolve over time. In order to bound these terms, we use Weyl's inequality (Lemma 3.3) to show that w.h.p. the gaps in the top $k + 1$ eigenvalues $\Delta_{ij}(t)$ satisfy $\Delta_{ij}(t) \geq \Omega(\sigma_i - \sigma_j)$ for every time $t \in [0, T]$ (Lemma 4.4),

---

[1]The theorem is stated for sub-Gaussian entries in terms of a constant $C$; this constant is $C = 2$ in the special case where the entries are $N(0,1)$ Gaussian.

provided that the initial gaps are sufficiently large (Assumption 2.1) (See Appendix L for a discussion on why we need this assumption for our proof to work).

5. **Step 5: Integrating the stochastic differential equation.** Next, we express the expected squared Frobenius distance $\mathbb{E}[\|\Psi(T) - \Psi(0)\|_F^2]$ as an integral $\|\Psi(T) - \Psi(0)\|_F^2] = \mathbb{E}\left[\left\|\int_0^T \mathrm{d}\Psi(t)\right\|_F^2\right]$. We then apply Itô's Lemma (Lemma 3.1) to obtain a formula for this integral. Roughly speaking, the formula we obtain (Lemma 4.5) is

$$\mathbb{E}\left[\|\Psi(T) - \Psi(0)\|_F^2\right] \approx \int_0^T \mathbb{E}\left[\sum_{i=1}^d \sum_{j\neq i} \frac{(\lambda_i - \lambda_j)^2}{\Delta_{ij}^2(t)}\right]\mathrm{d}t + T\int_0^T \mathbb{E}\left[\sum_{i=1}^d \left(\sum_{j\neq i} \frac{\lambda_i - \lambda_j}{\Delta_{ij}^2(t)}\right)^2\right]\mathrm{d}t \tag{5}$$

6. **Step 6: Completing the proof.** Plugging the bound $\Delta_{ij}(t) \geq \Omega(\sigma_i - \sigma_j)$ into (5), and noting that the first term on the r.h.s. of (5) is at least as large as the second term since $\sigma_i - \sigma_j \geq \sqrt{d}$, we obtain the bound in Theorem 2.2.

## 4.2 Step 3: Computing the stochastic derivative $\mathrm{d}\Psi(t)$

$\Psi(t)$ is itself a matrix-valued diffusion. We use the eigenvalue and eigenvector dynamics 3 and 4 together with Itô's Lemma (Lemma 3.1) to compute the Itô derivative of this diffusion. Towards this end, we first decompose the matrix $\Psi(t)$ as a sum of its eigenvectors: $\Psi(t) = \sum_{i=1}^d \lambda_i u_i(t) u_i^\top(t)$. Thus, we have

$$\mathrm{d}\Psi(t) = \sum_{i=1}^d \lambda_i \mathrm{d}(u_i(t) u_i^\top(t)). \tag{6}$$

We begin by computing the stochastic derivative $\mathrm{d}(u_i(t) u_i^\top(t))$ for each $i \in [d]$, by applying the formula for the derivative of $u_i(t)$ in (4), together with Itô's Lemma (Lemma 3.1):

**Lemma 4.1 (Stochastic derivative of $u_i(t) u_j^\top(t)$).** *For all $t \in [0, T]$, $\mathrm{d}(u_i(t) u_i^\top(t)) = \sum_{j\neq i} \frac{\mathrm{d}B_{ij}(t)}{\gamma_i(t) - \gamma_j(t)}(u_i(t) u_j^\top(t) + u_j(t) u_i^\top(t)) + \sum_{j\neq i} \frac{\mathrm{d}t}{(\gamma_i(t) - \gamma_j(t))^2}(u_i(t) u_i^\top(t) - u_j(t) u_j^\top(t)).$*

The proof is in Appendix A. Plugging Lemma 4.1 into (6), we get an expression for $\mathrm{d}\Psi(t)$:

**Lemma 4.2 (Stochastic derivative of $\Psi(t)$; see Appendix B for proof).** *For all $t \in [0, T]$ we have that $\mathrm{d}\Psi(t) = \frac{1}{2}\sum_{i=1}^d \sum_{j\neq i}(\lambda_i - \lambda_j)\frac{\mathrm{d}B_{ij}(t)}{\gamma_i(t) - \gamma_j(t)}(u_i(t) u_j^\top(t) + u_j(t) u_i^\top(t)) + \sum_{i=1}^d \sum_{j\neq i}(\lambda_i - \lambda_j)\frac{\mathrm{d}t}{(\gamma_i(t) - \gamma_j(t))^2}u_i(t) u_i^\top(t).$*

## 4.3 Step 4: Bounding the eigenvalue gaps

The derivative in Lemma 4.2 contains terms with magnitude proportional to the inverse of the eigenvalue gaps $\Delta_{ij}(t) := \gamma_i(t) - \gamma_j(t)$. To bound these terms, we would like to show that $\inf_{t\in[0,T]} \Delta_{ij}(t) \geq \Omega(\sigma_i - \sigma_j)$ for each $i < j \leq k+1$ with high probability. Towards this end, we first apply the spectral norm concentration bound for Gaussian random matrices (Lemma 3.2), which provides a high-probability bound for $\|B(t)\|_2$ at any time $t$, together with Doob's submartingale inequality, to show that the spectral norm of the matrix-valued Brownian motion $B(t)$ does not exceed $T\sqrt{d}$ at any time $t \in [0, T]$ w.h.p.:

**Lemma 4.3 (Spectral norm bound).** *For every $T > 0$, we have, $\mathbb{P}\left(\sup_{t\in[0,T]} \|B(t)\|_2 > 2T\sqrt{d} + \alpha\right) \leq 2\sqrt{\pi}e^{-\frac{1}{8}\frac{\alpha^2}{T^2}}.$*

The proof appears in Appendix C. Next, we use Lemma 4.3 to bound the eigenvalue gaps:

**Lemma 4.4 (Eigenvalue gap bound).** *Whenever $\gamma_i(0) - \gamma_{i+1}(0) \geq 4T\sqrt{d}$ for every $i \in S$ and $T > 0$ and some subset $S \subset [d-1]$, we have $\mathbb{P}\left(\bigcup_{i\in S}\left\{\inf_{t\in[0,T]} \gamma_i(t) - \gamma_{i+1}(t) < \frac{1}{2}(\gamma_i(0) - \gamma_{i+1}(0)) - \alpha)\right\}\right) \leq 2\sqrt{\pi}e^{-\frac{1}{32}\alpha^2}.$*

To prove Lemma 4.4, we plug Lemma 4.3 into Weyl's Inequality (Lemma 3.3), to show that

$$\gamma_i(t) - \gamma_{i+1}(t) \geq \sigma_i - \sigma_{i+1} - \|B(t)\|_2 \geq \Omega(\sigma_i - \sigma_{i+1} - T\sqrt{d}) \geq \Omega(\sigma_i - \sigma_{i+1}),$$

with high probability for each $i \leq k$ (Lemma 4.4). The last inequality holds since Assumption 2.1 ensures $\sigma_i - \sigma_{i+1} \geq \frac{1}{2}T\sqrt{d}$ for $i \leq k$. The full proof is in Appendix D.

### 4.4 Step 5: Integrating the stochastic differential equation

Next, we would like to integrate the derivative $\mathrm{d}\Psi(t)$ to obtain an expression for $\mathbb{E}[\|\Psi(T) - \Psi(0)\|_F^2]$, and to then plug in our high-probability bounds (Lemma 4.4) for the gaps $\Delta_{ij}(t)$. To allow us to later plug in these high-probability bounds after we integrate and take the expectation, we define a new diffusion process $Z_\eta(t)$ which has nearly the same stochastic differential equation as 4.2, except that each eigenvalue gap $\Delta_{ij}(t)$ is not permitted to become smaller than the value $\eta_{ij} = \frac{1}{4}(\sigma_i - \max(\sigma_j, \sigma_{k+1}))$ for each $i < j$.

Towards this end, fix any $\eta \in \mathbb{R}^{d \times d}$, define the following matrix-valued Itô diffusion $Z_\eta(t)$ via its Itô derivative $\mathrm{d}Z_\eta(t)$:

$$\mathrm{d}Z_\eta(t) := \frac{1}{2}\sum_{i=1}^d \sum_{j \neq i}|\lambda_i - \lambda_j|\frac{\mathrm{d}B_{ij}(t)}{\max(|\Delta_{ij}(t)|, \eta_{ij})}(u_i(t)u_j^\top(t) + u_j(t)u_i^\top(t))$$
$$+ \sum_{i=1}^d \sum_{j \neq i}(\lambda_i - \lambda_j)\frac{\mathrm{d}t}{\max(\Delta_{ij}^2(t), \eta_{ij}^2)}u_i(t)u_i^\top(t), \tag{7}$$

with initial condition $Z_\eta(0) := \Psi(0)$. Thus, $Z_\eta(t) = \Psi(0) + \int_0^t \mathrm{d}Z_\eta(s)$ for all $t \geq 0$. We then integrate $\mathrm{d}Z_\eta(t)$ over the time interval $[0, T]$, and apply Itô's Lemma (Lemma 3.1) to obtain an expression for the Frobenius norm of this integral:

**Lemma 4.5 (Frobenius distance integral).** *For any $T > 0$,* $\mathbb{E}\left[\|Z_\eta(T) - Z_\eta(0)\|_F^2\right] =$

$$2\int_0^T \mathbb{E}\left[\sum_{i=1}^d \sum_{j \neq i}\frac{(\lambda_i - \lambda_j)^2}{\max(\Delta_{ij}^2(t), \eta_{ij}^2)}\mathrm{d}t\right] + T\int_0^T \mathbb{E}\left[\sum_{i=1}^d\left(\sum_{j \neq i}\frac{\lambda_i - \lambda_j}{\max(\Delta_{ij}^2(t), \eta_{ij}^2)}\right)^2\right]\mathrm{d}t.$$

To prove Lemma 4.5, we write

$$Z_\eta(T) - Z_\eta(0) = \frac{1}{2}\int_0^T \sum_{i=1}^d \sum_{j \neq i}|\lambda_i - \lambda_j|\frac{\mathrm{d}B_{ij}(t)}{\max(|\Delta_{ij}(t)|, \eta_{ij})}(u_i(t)u_j^\top(t) + u_j(t)u_i^\top(t))$$

$$- \int_0^T \sum_{i=1}^d \sum_{j \neq i}(\lambda_i - \lambda_j)\frac{\mathrm{d}t}{\max(\Delta_{ij}^2(t), \eta_{ij}^2)}u_i(t)u_i^\top(t). \tag{8}$$

To compute the Frobenius norm of the first term on the r.h.s. of (8), we use Itô's Lemma (Lemma 3.1), with $X(t) := \int_0^t \sum_{i=1}^d \sum_{j \neq i}|\lambda_i - \lambda_j|\frac{\mathrm{d}B_{ij}(s)}{\max(|\Delta_{ij}(s)|, \eta_{ij})}(u_i(s)u_j^\top(s) + u_j(s)u_i^\top(s))$ and the function $f(X) := \|X\|_F^2 = \sum_{i=1}^d \sum_{j=1}^d X_{ij}^2$. By Itô's Lemma, we have

$$\mathbb{E}[\|X(T)\|_F^2 - \|X(0)\|_F^2] = \mathbb{E}[\frac{1}{2}\int_0^t \sum_{\ell,r}\sum_{\alpha,\beta}(\frac{\partial}{\partial X_{\alpha\beta}}f(X(t)))R_{(\ell r)(\alpha\beta)}(t)\mathrm{d}B_{\ell r}(t)]$$

$$+ \mathbb{E}\left[\frac{1}{2}\int_0^t \sum_{\ell,r}\sum_{i,j}\sum_{\alpha,\beta}\left(\frac{\partial^2}{\partial X_{ij}\partial X_{\alpha\beta}}f(X(t))\right)R_{(\ell r)(ij)}(t)R_{(\ell r)(\alpha\beta)}(t)\mathrm{d}t\right], \tag{9}$$

where $R_{(\ell r)(ij)}(t) := \left(\frac{|\lambda_i - \lambda_j|}{\max(|\Delta_{ij}(t)|, \eta_{ij})}(u_i(t)u_j^\top(t) + u_j(t)u_i^\top(t))\right)[\ell, r]$, and where we denote by either $H_{\ell r}$ or $H[\ell, r]$ the $(\ell, r)$'th entry of any matrix $H$.

The first term on the r.h.s. of (9) is equal to zero since $\mathrm{d}B_{\ell r}(s)$ is independent of both $X(t)$ and $R(t)$ for all $s \geq t$ and the time-integral of each Brownian motion increment $\mathrm{d}B_{\alpha\beta}(s)$ has zero mean. To compute the second term on the r.h.s. of (9), we use the fact that $\frac{\partial^2}{\partial X_{ij}\partial X_{\alpha\beta}}f(X)$ is equal to 2 for $i = j$ and 0 for $i \neq j$

To compute the Frobenius norm of the second term on the r.h.s. of (8), we use the Cauchy-Schwarz inequality. The full proof appears in Appendix E.

## 4.5 Step 6: Completing the proof

To complete the proof, we plug in the high-probability bounds on the eigenvalue gaps from Section 4.3 into Lemma 4.5. Since by Lemma 4.4 $\Delta_{ij}(t) \geq \frac{1}{2}(\sigma_i - \sigma_j)$ w.h.p. for each $i, j \leq k+1$, and $\eta_{ij} = \frac{1}{4}(\sigma_i - \max(\sigma_j, \sigma_{k+1}))$, we must also have that $Z_\eta(t) = \Psi(t)$ for all $t \in [0, T]$ w.h.p. Plugging in the high-probability bounds $\Delta_{ij}(t) \geq \frac{1}{2}(\sigma_i - \sigma_j)$ for each $i, j \geq k+1$, and noting that $\lambda_i - \lambda_j = 0$ for all $i, j > k$, we get that

$$\mathbb{E}\left[\|\hat{V}\Lambda\hat{V}^\top - V\Lambda V^\top\|_F^2\right] = \mathbb{E}\left[\|\Psi(T) - \Psi(0)\|_F^2\right]$$

$$\leq 2\int_0^T \mathbb{E}\left[\sum_{i=1}^d \sum_{j\neq i} \frac{(\lambda_i - \lambda_j)^2}{(\sigma_i - \sigma_j)^2}\right] \mathrm{d}t + T\int_0^T \mathbb{E}\left[\sum_{i=1}^d \left(\sum_{j\neq i} \frac{\lambda_i - \lambda_j}{(\sigma_i - \sigma_j)^2}\right)^2\right] \mathrm{d}t$$

$$\leq T\sum_{i=1}^k \sum_{j=i+1}^d \frac{(\lambda_i - \lambda_j)^2}{(\sigma_i - \max(\sigma_j, \sigma_{k+1}))^2} + T^2 \sum_{i=1}^k \left(\sum_{j=i+1}^d \frac{\lambda_i - \lambda_j}{(\sigma_i - \max(\sigma_j, \sigma_{k+1}))^2}\right)^2. \quad (10)$$

Since $(\sigma_i - \max(\sigma_j, \sigma_{k+1}) \geq \Omega(\sqrt{d})$ for all $i \leq k$ and $j \in [d]$, we can use the Cauchy-Schwarz inequality to show that the second term is (up to a factor of T) smaller than the first term: $\sum_{i=1}^k \left(\sum_{j=i+1}^d \frac{\lambda_i - \lambda_j}{(\sigma_i - \max(\sigma_j, \sigma_{k+1}))^2}\right)^2 \leq \sum_{i=1}^k \sum_{j=i+1}^d \frac{(\lambda_i - \lambda_j)^2}{(\sigma_i - \max(\sigma_j, \sigma_{k+1}))}$. Plugging $T = \frac{\sqrt{2\log(\frac{1.25}{\delta})}}{\varepsilon}$ into (10), we obtain the bound in Theorem 2.2. For the full proof of Theorem 2.2, see Appendix F.

## 5 Conclusion and Future Work

We present a new analysis of the Gaussian mechanism for a large class of symmetric matrix approximation problems, by viewing this mechanism as a Dyson Brownian motion initialized at the input matrix $M$. This viewpoint allows us to leverage the stochastic differential equations which govern the evolution of the eigenvalues and eigenvectors of Dyson Brownian motion to obtain new utility bounds for the Gaussian mechanism. To obtain our utility bounds, we show that the gaps $\Delta_{ij}(t)$ in the eigenvalues of the Dyson Brownian motion stay at least as large as the initial gap sizes (up to a constant factor), as long as the initial gaps in the top $k+1$ eigenvalues of the input matrix are $\geq \Omega(\sqrt{d})$ (Assumption 2.1).

While we observe that our assumption on the top-$k+1$ eigenvalue gaps holds on multiple real-world datasets, in practice one may need to apply differentially private matrix approximation on any matrix where the "effective rank" of the matrix is $k$— that is, on any matrix where the $k$'th eigenvalue gap $\sigma_k - \sigma_{k+1}$ is large— including on matrices where the gaps in the other eigenvalues may not be large and may even be zero. Unfortunately, for matrices with initial gaps in the top-$k$ eigenvalues smaller than $O(\sqrt{d})$, the gaps $\Delta_{ij}(t)$ in the eigenvalues of the Dyson Brownian motion become small enough that the expectation of the (inverse) second moment term $\frac{1}{\Delta_{ij}^2(t)}$ appearing in the Itô integral (Lemma 4.5) in our analysis may be very large or even infinite. Thus, the main question that remains open is whether one can obtain similar bounds on the utility for differentially private matrix approximation for any initial matrix $M$ where the $k$'th gap $\sigma_k - \sigma_{k+1}$ is large, without any assumption on the gaps between the other eigenvalues of $M$.

Finally, this paper analyzes a mechanism in differential privacy, which has many implications for preserving sensitive information of individuals. Thus, we believe our work will have positive societal impacts and do not foresee any negative impacts to society.

## Acknowledgments and Disclosure of Funding

This research was supported in part by NSF CCF-2104528 and CCF-2112665 awards.

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
