# OpenReview forum: "Re-Analyze Gauss: Bounds for Private Matrix Approximation via Dyson Brownian Motion"
_NeurIPS.cc/2022/Conference — NeurIPS 2022 Accept_

### Official Review · Reviewer_uAWo · 2022-07-09

**Rating:** 7
**Confidence:** 4
**Soundness:** 4 excellent
**Presentation:** 4 excellent
**Contribution:** 3 good

**Summary:**

This paper provides a new upper bound on the Frobenius-distance utility of the Gaussian mechanism for the general matrix approximation problem under differential privacy. The analysis utilize Dyson Brownian motion to track the evolution of the utility, which is, to my best knowledge, the first application of this mathematical tool.

**Questions:**

1. Can the author provide more examples and concrete justification  on the assumption 2.1?
2. The bound achieved in [15] is high probability bound while Theorem 2.2 is an expectation bound? Is it fair to direct compare these results?

**Limitations:**

The major limitation is the assumption seems to be too strong.

**Strengths And Weaknesses:**

Strengths:
Originality: This paper's contribution is original given that this is the first improvement of the utility bound of Gaussian mechanism. It is very interesting to see the application of stochastic differential equations together with dyson brownian motion to be applied in the proof.
Quality: I briefly checked the proof, it looks sound to my best knowledge.
Clarity: Overall the paper is well organized. The problem studied, existing results, the major contribution are clearly stated. I have no difficulty in reading the paper.
Significance: From my own perspective, I would value this improvement of the utility bound. First, the matrix recovery problem and its privacy problem are critical in applications such as recommender systems. Second, existing result is definitely not tight and thus an improvement should be expected.

Weakness:
Significance: My major concern is on the assumption 2.1. It requires very specific eigengap conditions. Though the paper proves that Wishart matrices  satisfy this condition. It would be better to provide more justifications and examples on the assumption.

---

> ### Author Response · Authors · 2022-08-02
> **Response to Reviewer uAWo**
>
> *Thank you for your valuable comments and suggestions. We are glad you find our contribution to be original and our application of Dyson Brownian Motion to be very interesting. We answer your specific questions below.*
>
> *``...provide more justifications and examples on the assumption."*
>
> Prior works oftentimes assume that the gap $\sigma_k - \sigma_{k+1}$ between the $k$'th and $k+1$'st eigenvalue is large  (e.g., many of the results in [Dwork, Talwar, Thakurta, Zhang, STOC 2014] assume $\sigma_k-\sigma_{k+1}\geq\sqrt{d}$). These assumptions are motivated in part by real-world datasets considered in prior works on differentially private matrix approximation and PCA. However, it is also true that standard datasets used in many of these prior works have large eigenvalue gaps $\sigma_i - \sigma_{i+1}$ between *all* of the top-$k$ eigenvalues, and that many of these datasets satisfy our Assumption 2.1.
>
> In Appendix J of the revised version of our paper, we consider three standard datasets from the UCI repository, the US Census dataset ($d=124$, $n= 2458285$ after standard pre-processing), KDD Cup dataset ($d=36$, $n = 494020$), and Adult dataset ($d=6$, $n= 48842$). All three of these datasets were previously used as benchmarks in the differentially private matrix approximation and PCA literature (Census and KDD Cup in [Chaudhuri, Sawarte, Sinha, NeurIPS 2012], and Adult dataset in e.g. [Amin, Dick, Kulesza, Munoz, Vassilvitskii, NeurIPS 2019]). We observe that, for values of $k$ large enough for a rank-$k$ approximation to recover 99% of the Frobenius norm of the covariance matrix, the gaps in the top-$k$ eigenvalues are sufficiently large for our Assumption 2.1 to hold for covariance matrices from all three of these datasets.
>
> For instance, for the covariance matrix $M$ of the Census dataset ($d=124$, $n= 2458285$) we observe that the top-$10$ eigenvalues account for 99% of the Frobenius norm of $M$, and that for any $k \leq 11$, the gaps in the top-$k$ eigenvalues of $M$ are sufficiently large to satisfy our Assumption 2.1 for  $\varepsilon =1$ and $\delta = \frac{1}{100}$  (Figure: see https://github.com/hcjc42zxai83/DBM/files/9244018/Gaps.pdf or Figure 3 in our revised paper). Namely, on this dataset our Assumption 2.1 requires that $\sigma_{i} - \sigma_{i+1} \geq 442$ for all $i \leq k$, and we observe that this condition holds for any $k \leq 11$.
>
> Finally, we note that, to allow one to verify Assumption 2.1 on real-world datasets, in the revised version of our paper we have replaced the universal constant in Assumption 2.1 with an explicit constant.
>
>
> *``high probability bound..."*
>
> While our current result holds in expectation, it is possible to use our techniques to prove high-probability bounds. We have added a discussion on this in Appendix M of our revised paper.
>
> The simplest approach is to plug in our expectation bound (Theorem 2.2) into Markov's inequality, which says that $P(|| \hat{V} \Lambda \hat{V}^\top -  V \Lambda V^\top ||_F^2 \geq s) \leq \frac{E(|| \hat{V} \Lambda \hat{V}^\top -  V \Lambda V^\top ||_F^2)}{s}$ for $s>0$.
>
> While Markov's inequality gives a high-probability bound, this bound decays as $\frac{1}{s}$. One approach to obtaining high-probability bounds which decay with rate exponential in $s$ might be to apply concentration inequalities to the part of our proof where we currently use expectation. Namely, our proof of Lemma 4.5 in the appendix uses Ito's Lemma to show that, roughly,
>
> $||\Psi(T)-\Psi(0)||_F^2$
>
>  $=4\int_0^T \sum_{i=1}^{d}\sum_{j \neq i}\frac{(\lambda_i-\lambda_j)^2}{(\gamma_i(t)-\gamma_j(t))^2}$
>
> $+\frac{1}{2}\int_0^t\sum_{\ell, r}\sum_{\alpha, \beta}\left(\frac{\partial}{\partial X_{\alpha \beta}}f(X(t))\right)R_{(\ell r)(\alpha \beta)}(t) \mathrm{d}B_{\ell r}(t),$
>
> where we define $X(t):=\int_0^{t}\sum_{i=1}^{d} \sum_{j \neq i} |\lambda_i-\lambda_j|\frac{\mathrm{d}B_{ij}(s)}{|\gamma_i(s)-\gamma_j(s) |}(u_i(s) u_j^\top(s)+u_j(s) u_i^\top(s))$, $R_{(\ell r) (i j)}(t) := \left(\frac{ |\lambda_i-\lambda_j| }{|\gamma_i(t)-\gamma_j(t)|}(u_i(t) u_j^\top(t)+u_j(t)u_i^\top(t)) \right)[\ell, r]$, and $f(X):= ||X||_F^2$.
>
>
> For simplicity, our current proof bounds the two integrals on the RHS by taking their expectation.  In particular, the second integral  vanishes as it has mean $0$.
>
> We do not have to do any additional work to bound the first term on the RHS w.h.p., since the only random variables appearing in that term are the eigenvalue gaps $\gamma_i(t) - \gamma_j(t)$,  and we already bound these gaps w.h.p. (Lemma 4.4). However, to bound the second integral w.h.p., we would also need to deal with the Gaussian random variables $B_{\ell r}(t)$ appearing inside the integral. One approach to bounding these random variables $B_{\ell r}(t)$ w.h.p. would be to apply standard Gaussian concentration inequalities, and it would be interesting to see whether this leads to high-probability bounds which are as tight as our current expectation bounds.

---

> > ### Comment · Reviewer_uAWo · 2022-08-08
> > **Thanks for the responses.**
> >
> > Thank you for these responses. My concern on Assumption 2.1 and expectation bound in Theorem 2.2 has been addressed.

---

### Official Review · Reviewer_utx8 · 2022-07-10

**Rating:** 3
**Confidence:** 4
**Soundness:** 2 fair
**Presentation:** 3 good
**Contribution:** 2 fair

**Summary:**

This paper discusses a matrix approximation in the context privacy using Dyson Brownian motion. By assuming a gap in eigenvalues, the authors established an error bound in privacy.

**Questions:**

The paper is well written and I don’t have questions.

**Strengths And Weaknesses:**

The paper is well-written, easy to understand, and scientifically correct. In addition, it is the first time I have seen that DBM is used to prove results in privacy setting.
However, I have two concerns. First the paper is not that relevant to ML community, and it is essentially a problem of matrix approximation, and privacy just provides some possible context. Second the results in the paper are conceptually easy, and not too profound (most based on well known properties of stochastic calculus and DBM).

---

> ### Author Response · Authors · 2022-08-02
> **Response to Reviewer utx8**
>
> *Thank you for taking the time to review our paper.  We hope you will consider increasing your score if your concerns are addressed.*
>
> *``relevant to ML community...privacy"*
>
> Differentially private (DP) matrix approximation problems have been widely studied and applied by the ML community. These include the problems of DP rank-$k$ matrix approximation which were previously studied, e.g., in [Dwork, Talwar, Thakurta, Zhang, STOC 2014], [Amin, Dick, Kulesza, Munoz, Vassilvitskii, NeurIPS 2019], [Arora,  Braverman, Upadhyay, NeurIPS 2018], and [Kapralov, Talwar, SODA 2013], and which we study in our Corollary 2.3. It also includes the problem of DP subspace recovery studied, e.g., in [Dwork, Talwar, Thakurta, Zhang, STOC 2014], and [Hardt, Price, NeurIPS 2014], and which we study in our Corollary 2.4.
>
> Moreover, while our paper provides tighter utility guarantees for a widely-used existing algorithm-- the Gaussian mechanism-- our guarantees can be used by practitioners when selecting the privacy parameters of this mechanism. This is because there is an inherent tradeoff between the privacy level and utility of DP mechanisms where stronger privacy parameters can lead to lower utility guarantees, and the privacy parameters $(\varepsilon, \delta)$ are chosen by practitioners as a compromise between privacy and utility [Dwork, McSherry, Nissim, Smith, 2006].
>
> Our results provide tighter utility bounds for DP covariance approximation and subspace recovery when the gaps in the top-$k$ eigenvalues are roughly $\Omega(\sqrt{d})$.  For instance, our Corollary 2.3 implies that the distance between the (non-private) rank-$k$ approximation $M_k$ of the covariance matrix $M$ and the rank-$k$ approximation $\hat{M}_k$ of the privatized matrix $\hat{M}$ obtained from the Gaussian mechanism with privacy parameters $(\varepsilon, \delta)$ satisfy the utility bound $\mathbb{E}[||\hat{M}_K-M_k ||_F ]\leq O(\sqrt{kd}\frac{\log^{\frac{1}{2}}}{\epsilon})$ whenever the gaps in the top-$k$ eigenvalues of $M$ are at least as large as $\Omega(\sqrt{d} \frac{\log^{\frac{1}{2}}\delta}{\epsilon})$ (Assumption 2.1), improving by a factor of $\sqrt{k}$ on the bound of roughly $\tilde{O}(k\sqrt{d})$ of [Dwork, Talwar, Thakurta, Zhang, STOC 2014] for datasets satisfying Assumption 2.1. Thus, our results can allow practitioners to set stronger privacy parameters for the Gaussian mechanism with a better tradeoff between privacy and utility for covariance matrix approximation whenever (Assumption 2.1) is satisfied.
>
> *Relevance of our techniques to ML community beyond privacy:* To obtain our tighter analysis of the Gaussian mechanism, our paper proves new matrix perturbation bounds, which are tighter than previous bounds in settings when the perturbation is a random matrix. Namely, given a covariance matrix $M$, our Theorem 2.2 proves tighter perturbation bounds on rank-$k$ approximations to the perturbation $\hat{M}=M+E$ for perturbations $E = G+G^\top$ where $G$ is a random matrix with iid $N(0, \frac{\log \delta}{\varepsilon^2})$ entries. For instance, in another corollary to our main result (Corollary 2.4) (setting $\epsilon, \delta = \Theta(1)$ for simplicity), we show that the distance between projection matrices $V_k V_k^\top$ and $
> \hat{V}_k\hat{V}_k^\top$ onto the subspaces spanned by the top-$k$ eigenvectors of $M$ and $\hat{M}$, respectively, satisfies
>
> $ \mathbb{E}[||\hat{V}_k\hat{V}_k^\top-V_k V_k^\top||_F]$
>
> $ \leq O(\frac{\sqrt{d}}{\sigma_{k}-\sigma_{k+1}}),$
>
> where $\sigma_k$ and $\sigma_{k+1}$ are the $k$'th and $k+1$'st eigenvalues of $M$, whenever the gaps in the top-$k$ eigenvalues  satisfy $\sigma_i - \sigma_{i+1} \geq \Omega(\sigma_k - \sigma_{k+1}) \geq \Omega(\sqrt{d})$ for $i \leq k$. This improves on the ``worst-case'' eigenvector perturbation bounds of [Davis, Kahan, 1970], in the above setting where the perturbation is a random matrix. Roughly, the Davis-Kahan perturbation theorem states that for any (possibly non-random) perturbation $E$,
>
> $||\hat{V}_k\hat{V}_k^\top-V_k V_k^\top||_2\leq\frac{||E|| _2}{\sigma_k-\sigma_k+1}.$
>
> In the special case where the perturbation is a Gaussian random matrix, this implies
>
> $||\hat{V}_k\hat{V}_k^\top-V_k V_k^\top||_F$
>
> $\leq\frac{\sqrt{kd}}{\sigma_k-\sigma_{k+1}}$
>
> w.h.p. (see e.g. [Dwork, Talwar, Thakurta, Zhang, STOC 2014]). Thus, we obtain a bound which is tighter (in expectation) by a factor of $\sqrt{k}$ than the bound implied by the Davis-Kahan perturbation theorem in the special case where the perturbation is a random matrix satisfying the above eigenvalue gap condition.
>
> In addition to applications to DP, Matrix perturbation bounds such as the Davis-Kahan theorem have many other applications to ML, e.g., they are used to prove generalization bounds in statistical learning [Yu, Wang, Samworth, Biometrika 2015]. Thus, our matrix perturbation results may also be applicable to areas of ML beyond privacy, in settings where the data is perturbed by random matrix noise.

---

### Official Review · Reviewer_RW4L · 2022-07-12

**Rating:** 6
**Confidence:** 3
**Soundness:** 3 good
**Presentation:** 3 good
**Contribution:** 3 good

**Summary:**

This paper gives a new analysis of the Gaussian noise mechanism for approximating covariances matrices with differential privacy. The basic mechanism is quite simple: given a private covariance matrix $M$, add a Gaussian noise matrix $E$ to form $\hat{M} = M + E$. For an appropriate choice of the variance of the Gaussian, $\hat{M}$ preserves differential privacy. Then one can use post-processing to compute some function of $f(\hat{M})$ and hope that this approximates $f(M)$. For example, $f( . )$ could be projection on the span of the top $k$ eigenvalues, or the best rank $k$ approximation.

The performance of the Gaussian noise mechanism for this class of problems was analyzed in an important prior paper of Dwork, Talwar, Thakurta, and Zhang. They also showed that, for some problems, the mechanism is worst-case optimal, and gave improved guarantees when the input matrix satisfies some spectral assumptions (large gaps between eigenvalues).

The present paper gives a different analysis of the Gaussian noise mechanism, based on viewing the $\hat{M}$ as the value of a matrix-valued Brownian motion at some time $T$, where the Brownian motion starts at $M$. This induces an Ito process on the eigenvalues and eigenvectors of the matrix, and allows using Ito calculus to bound the error of the Gaussian noise mechanism for problems of the type described above. Generally, the bounds are better than prior work, but, as far as I understand, require stricter assumptions on the spectrum of $M$.

**Questions:**

Can you give an argument why the stochastic calculus techniques give results that could not be proved with the techniques of [15], even under the stronger assumptions made in this paper? An informal argument would also be helpful.

How can one use your analysis to give error bounds for covariance approximation, given that the eigenvalues of $M$ are private information and should not be used in post-processing without adding noise to them?

**Limitations:**

I don’t have concerns.

**Strengths And Weaknesses:**

The new method of analysis via stochastic calculus is, in my opinion, the strongest point about this paper. This is a new approach, and may prove fruitful for other problems, or for deriving tighter error bounds.

On the other hand, the results themselves are hard to compare with prior work, since they require significantly stronger spectral assumptions. The authors also don’t clarify why these types of results cannot be proved using the techniques of reference [15]. This makes it hard to appreciate what is gained by adopting the machinery of stochastic calculus. It would be nice to have at least one application where the new analysis gives better results than prior work, without making any additional assumptions.

I also don’t understand how the results in this paper can be used for covariance approximation. The algorithm seems to require public knowledge of the eigenvalues of the input covariance matrix, but this is private information. The authors make one cryptic comment about this, but I still don’t understand how their framework can be used for this problem to get an actual differentially private algorithm.

---

> ### Author Response · Authors · 2022-08-02
> **Part II of Response to Reviewer RW4L**
>
> *Thank you for your valuable comments and suggestions. We are encouraged you find our method of analysis to be a new and fruitful approach. We answer your questions below and we hope you will consider increasing your support if we have addressed them.*
>
> Due to space constraints, we answer your questions in two different posts.   In this first post we answered the first two questions and comments about (1) our spectral assumptions and (2) "why these types of results cannot be proved using the techniques of reference [15]...".  In this post we answer your question (3) ``eigenvalues of M... should not be used in post-processing''
>
> *3) ``eigenvalues of M... should not be used in post-processing''*
>
> Sorry for the confusion--perhaps our statement needs more explanation. The algorithm in our Corollary 2.3 is private because it uses eigenvalues from the perturbed matrix $\hat{M}$ for post-processing, not from the original matrix $M$. We added a detailed explanation of why this ensures privacy in the revised version of our paper (Lines 739-761 in proof of Corollary 2.3).
>
> More specifically, the steps of the algorithm considered in Corollary 2.3  are:
>
> 1. Add Gaussian noise $G$ to $M$ to obtain a perturbed matrix $\hat{M}=M+G+G^\top$,
>
> 2. Compute the spectral decomposition $\hat{M}= \hat{V}\hat{\Sigma}\hat{V}^\top$,
>
> 3. Take the top-$k$ eigenvalues $\hat{\sigma}_1, \ldots,\hat{\sigma}_k$ of  $\hat{M}$, and set $\hat{\Sigma}_k=\mathrm{diag}(\hat{\sigma}_1, \ldots,\hat{\sigma}_k,0,\ldots,0)$,
>
> 4. Output $\hat{M}_k := \hat{V}\hat{\Sigma}_k \hat{V}^\top$.
>
> Step (1) is just the Gaussian mechanism, which is $(\varepsilon, \delta)$-DP (e.g., by Theorem A.1 of [Dwork, Roth, 2014]). And Steps (2), (3), (4) do not require additional dataset access, and therefore preserve privacy. In particular, the eigenvalues $\hat{\sigma}_1,\ldots,\hat{\sigma}_k$ of $\hat{M}_k$ are obtained from the perturbed matrix $\hat{M}$, and thus do not compromise privacy.
>
> To bound the utility of the rank-$k$ approximation $\hat{M}_k = \hat{V}\hat{\Sigma}_k\hat{V}^\top$ which is the output of the above algorithm, we first  apply our Theorem 2.2 to bound  $E[||\hat{V}\Sigma_k\hat{V}^\top-V \Sigma_k V^\top||_F^2]$ (Eq. 29, in either version of our paper). We then use Weyl's Inequality to bound the distance $|\hat{\sigma}_i - \sigma_i|$ between the original eigenvalues and the perturbed eigenvalues, and use this to bound the quantity $E[||\hat{V}\hat{\Sigma}_k \hat{V}^\top -  \hat{V}\Sigma_k \hat{V}^\top ||_F^2]$ (Eq. 31). Combining these two bounds, we get a bound on the utility  $E[||\hat{V}\hat{\Sigma}_k \hat{V}^\top -  \hat{V}\Sigma_k \hat{V}^\top||_F^2]$ of the output $\hat{M}_k = \hat{V}\hat{\Sigma}_k\hat{V}^\top$ (Eq. 32).

---

> > ### Comment · Reviewer_RW4L · 2022-08-05
> > **Thanks for clarifications**
> >
> > Thank you for these helpful responses - I think my main concerns about correctness and the role of the Brownian motion analysis are addressed.

---

> ### Author Response · Authors · 2022-08-02
> **Part I of Response to Reviewer RW4L**
>
> *Thank you for your valuable comments and suggestions. We are encouraged you find our method of analysis to be a new and fruitful approach. We answer your questions below and we hope you will consider increasing your support if we have addressed them.*
>
> Due to space constraints, we answer your questions in two different posts.  In this first post we answer the first two questions about (1) our spectral assumptions and (2) "why these types of results cannot be proved using the techniques of reference [15]...".  In the next post we answer your question (3) ``eigenvalues of M... should not be used in post-processing''
>
> *1) ``...spectral assumptions"*
>
>  Prior works oftentimes assume the $k$'th eigenvalue gap $\sigma_k - \sigma_{k+1}$ is large  (e.g., many results in [Dwork, Talwar, Thakurta, Zhang, STOC 2014] ). However, standard datasets used in prior works also have large gaps $\sigma_i-\sigma_{i+1}$ between all the top-$k$ eigenvalues, and many of these satisfy our Assumption 2.1. In Appendix J of our revised paper, we consider three standard datasets from the UCI repository, the Census dataset ($d=124$, $n= 2458285$ after standard pre-processing), KDD Cup dataset ($d=36$, $n = 494020$), and Adult dataset ($d=6$, $n= 48842$). All three were used as benchmarks in the DP matrix approximation literature (Census and KDD Cup in [Chaudhuri, Sawarte, Sinha, 2012], Adult in [Amin, Dick, Kulesza, Munoz, Vassilvitskii, 2019]). On all three datasets we observe that, for values of $k$ large enough for a rank-$k$ approximation to recover 99% of the Frobenius norm of the covariance matrix $M,$ the gaps in the top-$k$ eigenvalues are sufficiently large for our Assumption 2.1 to hold.
>
> For instance, for Census dataset we observe the top-$10$ eigenvalues account for 99% of the Frobenius norm. And Assumption 2.1 (for $\varepsilon=1$ and $\delta=\frac{1}{100}$) requires  $\sigma_{i}-\sigma_{i+1} \geq 442$ for $i \leq k$; we observe this condition holds for any $k\leq 11$ on this dataset (Figure: see https://github.com/hcjc42zxai83/DBM/files/9244018/Gaps.pdf or Fig. 3 in revised paper).
>
> *2) ``why these types of results cannot be proved using the techniques of reference [15]..."*
>
> Previous techniques used in [15] indeed do not lead to the bounds we obtain; we have added a section to our revised paper (Appendix K) discussing these difficulties. To see why, for covariance approximation, [15] use the  trace inequality $\mathrm{tr}(X) \leq \mathrm{rank}(X)||X||_2$  for $X \in \mathbb{R}^{d \times d}$, to show $||M-\hat{V}_k\hat{\Sigma}_k \hat{V}_k^\top||_F - ||M-V_k\Sigma_k V_k^\top||_F\leq O(k\sqrt{d})$ w.h.p.  For $k=d$, this bound is $O(d^{1.5})$, and thus is not tight since $||M-\hat{V}_k\hat{\Sigma}_k\hat{V}_k^\top||_F-||M-V_k\Sigma_k V_k^\top||_F=O(d)$ w.h.p. Roughly, the additional $\sqrt{k} =\sqrt{d}$ factor is because the trace inequality uses the spectral norm, even though they only need a bound in Frobenius norm.
>
> An alternative approach, taken by [15] to prove bounds for subspace recovery, is to use the eigenvector perturbation theorem of [Davis, Kahan, 1970], which, when applied to the Gaussian mechanism, implies, roughly,  $|| \hat{V}_k \hat{V}_k^{\top} -V_k V_k^{\top}||_F  \leq$
>   $\frac{\sqrt{k}\sqrt{d}}{\sigma_k - \sigma_k+1}$  w.h.p.
> To apply Davis-Kahan to covariance approximation, we could write, roughly
>
> $||\hat{V} \Sigma_k \hat{V}^\top - V \Sigma_k V^\top||_F$
>
>  $\leq \sum_{i=1}^{k-1} (\sigma_i - \sigma_{i+1}) ||\hat{V}_i \hat{V}_i^\top -V_i  V_i^\top||_F$
>
> $ \leq \sum_{i=1}^{k-1}(\sigma_i-\sigma_{i+1}) \frac{\sqrt{i} \sqrt{d}}{\sigma_i-\sigma_i+1} = O(k^{1.5}\sqrt{d})$.
>
> As a first step to obtain a tighter bound, we would ideally like to add up the terms $||\hat{V}_i \hat{V}_i^\top - V_i V_i^\top||_F$ as a sum-of-squares. However, this requires bounding the cross-terms $\mathrm{tr} ((\hat{V}_i \hat{V}_i^\top - V_i V_i^\top)   (\hat{V}_j \hat{V}_j^\top - V_j V_j^\top))$.
>
> Roughly, we handle such cross-terms by viewing the addition of noise as a continuous-time matrix diffusion $\Psi(t) = M + B(t)$, whose eigenvalues $\gamma_i(t)$ and eigenvectors $u_i(t)$, $i\in [d]$, evolve over time. This allows us to ``add up'' contributions of  different eigenvectors to the Frobenius distance as a stochastic integral,
>
> $||\hat{V}\Sigma_k \hat{V}^\top-V \Sigma_k V^\top||_F^2$
>
> $= ||\int_0^{T}\sum_{i=1}^{d}\sum_{j \neq i} |\sigma_i-\sigma_j|\frac{\mathrm{d}B_{ij}(t)}{\gamma_i(t)-\gamma_j(t)}(u_i(t) u_j^\top(t)+u_j(t)u_i^\top(t))||_F^2,$
>
> where, roughly, each differential cross term  $\frac{\mathrm{d}B_{ij}(t)}{\gamma_i(t)-\gamma_j(t)}(u_i(t)u_j^\top(t)+u_j(t)u_i^\top(t))$ adds noise to the matrix $V\Sigma_k V^\top$ independently of the other terms since the Brownian motion differentials $\mathrm{d}B_{ij}(t)$ are independent for all $i,j,t$. (Eq. 15, either version of our paper). Roughly, this allows us to add up the contributions of these terms as a sum of squares using Ito's Lemma from stochastic calculus (Eq. 18).

---

### Official Review · Reviewer_6MaL · 2022-07-14

**Rating:** 6
**Confidence:** 3
**Soundness:** 3 good
**Presentation:** 4 excellent
**Contribution:** 3 good

**Summary:**

The authors consider the problem where, given a symmetric matrix $M$ and a vector of eigenvalues $\lambda$, we want to privately estimate a matrix that shares the same set of orthogonal unit eigenvectors as $M$ and has eigenvalues $\lambda$. They obtain new bounds for this problem thanks to a novel interpretation of the Gaussian noise as a Dyson Brownian motion. They apply their results to obtain bounds for covariance estimation, low-rank approximation and subspace recovery.

**Questions:**

The thing I mainly want to ask from the authors it to justify Assumption 2.1 in the context of the problems they are considering e.g., by giving pointers to other papers with similar assumptions that are from the covariance estimation/PCA literature. Currently, I am leaning towards rejecting the paper, but I am willing to raise my score if the authors can convince me that their assumption is justified within the context of the problems considered and has not simply been imported from SDEs/random matrix theory.

=== UPDATE ===

After taking into account the rebuttal and discussion with other reviewers and the AC, I decided to raise my score from 4 to 6.

**Limitations:**

This is a purely theoretical paper, so it doesn't have an obvious societal impact.

**Strengths And Weaknesses:**

Strength: The interpretation of the Gaussian mechanism as an SDE is novel to the best of my knowledge and the way it is leveraged to obtain new bounds for these classic problems in private statistics is interesting.

Weakness: My main issue with the paper is that I feel that Assumption 2.1 is contrived. The authors argue that this condition is satisfied whp by Wishart matrices. However, other works I've read e.g., on the topics of PCA or subspace recovery assume that we have a range of eigenvalues (say the top $k$ eigenvalues) which are similar in magnitude, whereas there are gaps between them and the smaller eigenvalues. Judging based on that, I feel it's unnatural to assume the existence of additive gaps among the ``large" eigenvalues.

Comment: I have not gone through the proofs in the appendix in detail, because I am not too familiar with SDEs, so I can't really follow their low-level aspects. However, the high level picture described in the main body seems reasonable and I couldn't find any obvious faults.

---

> ### Author Response · Authors · 2022-08-02
> **Response to Reviewer 6MaL**
>
> *Thank you for your valuable comments and suggestions. We are encouraged that you find our interpretation of the Gaussian mechanism as an SDE to be novel and interesting. We answer your specific questions below and we thank you for offering to increase your score if your concerns are addressed.*
>
> ``justify Assumption 2.1 in the context of the problems they are considering..."
>
> Prior works oftentimes assume that the gap $\sigma_k - \sigma_{k+1}$ between the $k$'th and $k+1$'st eigenvalue is large  (e.g., many of the results in [Dwork, Talwar, Thakurta, Zhang, STOC 2014] assume that $\sigma_k - \sigma_{k+1} \geq \sqrt{d}$). These assumptions are motivated in part by real-world datasets considered in prior works on differentially private matrix approximation and PCA. However, it is also true that standard datasets used in many of these prior works have large eigenvalue gaps $\sigma_i - \sigma_{i+1}$ between *all* of the top-$k$ eigenvalues, and that many of these datasets satisfy our Assumption 2.1.
>
> In Appendix J of the revised version of our paper, we consider three standard datasets from the UCI repository, the US Census dataset ($d=124$, $n= 2458285$ after standard pre-processing), KDD Cup dataset ($d=36$, $n = 494020$), and Adult dataset ($d=6$, $n= 48842$). All three of these datasets were previously used as benchmarks in the differentially private matrix approximation and PCA literature (Census and KDD Cup in [Chaudhuri, Sawarte, Sinha, NeurIPS 2012], and the Adult dataset in e.g. [Amin, Dick, Kulesza, Munoz, Vassilvitskii, NeurIPS 2019]). We observe that, for values of $k$ large enough for a rank-$k$ approximation to recover 99% of the Frobenius norm of the covariance matrix, the gaps in the top-$k$ eigenvalues are sufficiently large for our Assumption 2.1 to hold for covariance matrices from all three of these datasets.
>
> For instance, for the covariance matrix $M$ of the Census dataset ($d=124$, $n= 2458285$) we observe that the top-$10$ eigenvalues account for 99% of the Frobenius norm of $M$, and that for any $k \leq 11$, the gaps in the top-$k$ eigenvalues of $M$ are sufficiently large to satisfy our Assumption 2.1 for  $\varepsilon =1$ and $\delta = \frac{1}{100}$  (Figure: see https://github.com/hcjc42zxai83/DBM/files/9244018/Gaps.pdf or Figure 3 in our revised paper). Namely, on this dataset our Assumption 2.1 requires that $\sigma_{i} - \sigma_{i+1} \geq 442$ for all $i \leq k$, and we observe that this condition holds for any $k \leq 11$.
>
> Finally, we note that, to allow one to verify Assumption 2.1 on real-world datasets, in the revised version of our paper we have replaced the universal constant in Assumption 2.1 with an explicit constant.

---

> > ### Comment · Reviewer_6MaL · 2022-08-08
> > **Reviewer Response**
> >
> > Thank you for your response. As indicated in my review, I was hoping for theoretical justification with pointers to other works in the literature with similar assumptions. However, arguing about the merits of the guarantees in the paper using experiments on real world datasets is something that I am willing to taking into consideration as well. That said, I'd like to wait for the next phase of the discussion period (where reviewers and metareviewers discuss the papers) before potentially changing my score.

---

### Meta-Review · Area_Chair_fy4a · 2022-08-23

**Recommendation:** Accept
**Confidence:** Less certain

**Metareview:**

This paper brings new mathematical tools (Dyson Brownian motion) to analyze the utility of a new mechanism, which improves over existing techniques when certain conditions on the original matrix spectrum is met. Although the conditions can be restrictive, the novelty of the idea can open new doors to differential private mechanism design.

**Award:**

No

---

### Decision · Program_Chairs · 2022-09-14

Accept